# Alternative Splicing-Mediated Resistance to Antibody-Based Therapies: Mechanisms and Emerging Therapeutic Strategies

**DOI:** 10.3390/ijms262411918

**Published:** 2025-12-10

**Authors:** Sanga Choi, Jieun Kang, Jung-Hyun Kim

**Affiliations:** Research Institute, National Cancer Center, 323 Ilsan-ro, Goyang-si 10408, Gyeonggi-do, Republic of Korea

**Keywords:** alternative splicing, antibody therapy resistance, immune checkpoint blockade, splicing factors, precision oncology, isoform diversity

## Abstract

Antibody-based therapeutics targeting tumor surface markers have transformed cancer treatment; however, their efficacy is frequently limited by tumor escape mechanisms such as antigen loss, phenotypic switching, and heterogeneous target expression. Beyond genetic or transcriptional changes, RNA alternative splicing (AS) has emerged as a central post-transcriptional mechanism driving antigenic diversity and immune escape. This review outlines how AS-generated isoforms remodel surface antigen structure and function across key therapeutic targets—including CD/19/CD20/CD22, EGFR/HER2, VEGF, and PD-1/PD-L1—thereby promoting resistance to monoclonal antibodies, antibody–drug conjugates, and immune checkpoint inhibitors. The aberrant activity of splicing regulators disrupts canonical exon selection, leading to altered receptor signaling or the secretion of soluble decoy isoforms that evade immune recognition. Emerging therapeutic strategies aim to counteract these processes through antisense oligonucleotide-mediated splicing correction, pharmacologic modulation of splicing regulators, and isoform-selective antibody or CAR-T designs. Collectively, understanding splicing-driven antigenic plasticity reveals an additional, dynamic layer of resistance regulation and provides a framework for developing RNA-informed precision antibody therapies designed to restore antigen expression, overcome immune escape, and enhance durable clinical responses.

## 1. Introduction

RNA splicing is a fundamental post-transcriptional process in which introns are precisely removed and exons are joined to generate mature messenger (mRNA) transcripts [1,2,3]. In addition to ensuring transcript fidelity, RNA splicing serves as a major driver of molecular diversity. Through alternative splicing (AS), a single gene can give rise to multiple coding and noncoding isoforms with distinct structural and functional properties, thereby expanding the diversity and adaptability of the cellular proteome [4,5,6].

Most multi-exon genes undergo AS, enabling cells to modulate protein interaction, subcellular localization, and signaling in a context-dependent manner. This regulatory flexibility supports growth, differentiation, and tissue homeostasis, whereas its dysregulation has been increasingly recognized as a major contributor to human disease [1,2,3,5].

RNA splicing is not a static process. It is dynamically regulated by splicing factors, auxiliary RNA-binding proteins, and cis-acting regulatory elements that respond to developmental programs, cellular stress, metabolic states, and microenvironmental signals [7,8,9,10]. This multilayered regulatory network allows rapid, reversible, and mutation-independent tuning of gene expression. When this regulation is disrupted, aberrant splicing can impair normal protein production or generate dysfunctional isoforms that drive pathogenesis.

RNA splicing defects contribute to a broad spectrum of human disorders, including spinal muscular atrophy (SMA), amyotrophic lateral sclerosis (ALS), Alzheimer’s disease, cardiovascular disorders, and immune dysregulation [11,12,13]. In these conditions, exon skipping, intron retention, or mutations in regulatory elements disrupt the balance of functional isoforms. Among these diseases, cancer is distinguished by especially pervasive and functionally significant splicing dysregulation. Tumor cells often reprogram their splicing machinery in response to oncogenic signals and microenvironmental stress, leading to the systematic production of isoforms that reshape cellular behavior. The next section outlines the major mechanisms through which RNA splicing becomes aberrantly regulated in cancer.

## 2. Mechanism of Aberrant Regulation of RNA Splicing in Cancer

AS becomes profoundly dysregulated in malignancy, with patterns that vary by tumor type, cellular lineage, microenvironment, and genetic alterations. These splicing abnormalities contribute directly to hallmarks of cancer, including uncontrolled proliferation, malignant transformation, immune evasion, and therapeutic resistance [7,14]. Three major mechanistic routes underlie this dysregulation.

(1)
**Mutations in core spliceosomal components**


Recurrent mutations in spliceosomal genes—*SF3B1*, *SRSF2*, *U2AF1*, and *ZRSR2*—are frequently observed across various cancer types, disrupting early steps of spliceosome assembly and splice-site recognition [9,15,16]. SF3B1, a key component of the U2 snRNP complex, harbors recurrent mutations that impair branch point site (BPS) recognition and promote usage of cryptic 3′ splice sites. Mutations in SRSF2 and U2AF1 alter RNA-binding specificity at 3′ splice sites, resulting in extensive mis-splicing patterns [9,15,16,17,18] (Figure 1A).

(2)
**Altered expression of splicing factors**


Oncogenic signaling and cellular stress frequently remodel the expression of serine/arginine-rich (SR) proteins and heterogeneous nuclear ribonucleoproteins (hnRNPs), producing cancer-specific AS patterns. SRSF1 overexpression enhances production of anti-apoptotic and pro-proliferative isoforms in breast, lung, and colorectal cancers [19,20,21]. Likewise, elevated SRSF3 expression drives preferential inclusion of exon 10 in *PKM* transcripts, promoting the cancer-associated PKM2 isoform that enhances glycolytic flux and tumor progression [21,22,23] (Figure 1B).

(3)
**Cis-acting mutations within RNA regulatory elements**


Point mutations or indels in splice sites, branch points, polypyrimidine tracts, or enhancer/silencer motifs disrupt recruitment of spliceosomal components. Such lesions generate aberrant, often tumor-specific isoforms—some with gain-of-function oncogenic activity, others producing neoantigens that reshape tumor immunogenicity [24,25] (Figure 1C).

Together, these disruptions reshape the splicing machinery of cancer cells and generate noncanonical mRNA isoforms with altered structural and functional properties. These aberrant splice variants are not passive byproducts of dysregulated splicing; rather, they actively drive malignant phenotypes. The following section describes how such oncogenic splice isoforms contribute to tumor initiation, proliferation, survival, invasion, and metastasis.

## 3. Oncogenic Functions of Aberrant Alternative Splicing Isoforms

Aberrant regulation of alternative splicing gives rise to isoforms that often possess biological activities distinct from—or even antagonistic to—their wild-type counterparts. Through these altered molecular properties, cancer-associated splice variants contribute to core malignant phenotypes, including uncontrolled proliferation, apoptotic resistance, metastatic dissemination, and angiogenic activation [26,27,28].

### 3.1. Tumor Growth and Proliferation

AS frequently rewires metabolic and growth-related pathways to favor tumor progression. The *PKM* gene undergoes mutually exclusive splicing of exons 9 and 10 to produce two isoforms, PKM1 and PKM2. Cancer cells preferentially express PKM2, which enhances glycolysis and the supply of biosynthetic precursors, thereby reinforcing the Warburg effect and enhancing proliferative capacity [29,30,31]. Elevated PKM2 expression correlates with poor clinical outcomes in colorectal cancer [31]. The tumor suppressor KLF6 similarly generates an oncogenic splice variant, KLF6-SV1, which antagonizes wild-type KLF6, promoting tumor growth and leading to unfavorable clinical outcomes across multiple cancers [32]. Additionally, the *RPS6KB1* gene produces a truncated isoform, RPS6KB1-2, that aberrantly activates mTORC1 signaling and drives growth in breast, lung, and other primary tumors [33] (Figure 1D).

### 3.2. Apoptosis Evasion (Resisting Cell Death)

Cancer cells exploit splicing to shift the balance between pro- and anti-apoptotic isoforms. The *BCL2L1* gene produces two major isoforms, anti-apoptotic Bcl-xL and pro-apoptotic Bcl-xS. Cancer cells preferentially favor Bcl-xL expression through splicing factor-mediated exon selection, promoting therapy resistance [34,35,36]. The *MCL1* gene similarly yields a long anti-apoptotic isoform (MCL1L) and a short pro-apoptotic isoform (MCL1S) [37]. SRSF1- and SRSF5-mediated splicing promotes MCL1L expression, conferring cancer cells with resistance to chemotherapeutic agents that induce apoptosis [37,38]. In colorectal cancer, SRSF7 promotes exon 6 skipping in *FAS*, resulting in a soluble Fas isoform that lacks the transmembrane domain. This truncated form acts as a decoy receptor, suppressing Fas-mediated apoptosis and further enhancing tumor cell survival [39] (Figure 1E).

### 3.3. Invasion and Metastasis

The AS of adhesion and signaling molecules promotes epithelial–mesenchymal transition (EMT), motility, and metastatic competence. The *CD44* gene undergoes extensive splicing to generate variant exon-containing isoforms (CD44v), which enhance EMT, invasiveness, and metastatic potential in breast and colorectal cancers [20,40,41,42]. ESRP1-mediated splicing drives CD44v isoform expression that facilitates lung metastasis in breast cancer [41]. Similarly, the *CEACAM1* gene produces two major isoforms via alternative exon 7 inclusion. The short isoform, CEACAM1-S, promotes proliferation, migration, and invasion in colorectal cancer and is inversely correlated with recurrence-free survival [43] (Figure 1F).

### 3.4. Tumor Angiogenesis

VEGFA undergoes AS to produce pro-angiogenic isoforms (e.g., VEGF_Axxx and VEGF165a) and anti-angiogenic isoforms (e.g., VEGF_Axxxb and VEGF165b). Tumors typically shift splicing toward pro-angiogenic variants, enhancing neovascularization and tumor progression [44,45,46]. Its receptor VEGFR2 also exists as a membrane-bound form (mVEGFR2) and a soluble inhibitory decoy form (sVEGFR2) generated through AS [17,47]. Alterations in these isoform ratios significantly influence the angiogenic balance within the tumor microenvironment (Figure 1G).

In summary, these aberrant splicing events extensively remodel the isoform landscape of cancer cells, enhancing proliferation, apoptosis resistance, metastasis, and angiogenesis. Importantly, many of these pathogenic isoforms modify surface or secreted proteins targeted by therapeutic antibodies, thereby directly linking splicing dysregulation to resistance mechanisms addressed in the following section.

## 4. Antibody-Based Therapeutics and the Impact of RNA Splicing on Therapeutic Resistance

Monoclonal antibody (mAb) therapeutics have emerged as one of the most precise and adaptable biologic treatment platforms [48,49,50,51]. Their high molecular specificity minimizes off-target cytotoxicity and enables the selective targeting of disease-associated antigens [51,52]. This therapeutic modality was initially developed for the treatment of oncological and immune-mediated disorders; however, its indications have now expanded to infectious, hematologic, neurologic, and metabolic disorders [52,53].

The mechanisms of action can be broadly categorized into (i) blockade of receptor–ligand interactions to inhibit oncogenic or immunosuppressive signaling [52], and (ii) activation of immune effector mechanisms, including antibody-dependent cellular cytotoxicity (ADCC), antibody-dependent cellular phagocytosis (ADCP), and complement-dependent cytotoxicity (CDC) [54]. Given their specificity and broad clinical utility, antibody-based therapies now constitute a central component of treatment across both hematologic malignancies and solid tumors [50,51,52,53].

Despite their efficacy, clinical resistance—or non-responsiveness—remains a significant clinical challenge. Many resistant tumors exhibit structural or quantitative alterations in target antigens, and mounting evidence implicates aberrant RNA splicing as a pivotal driver of these changes. Splicing dysregulation can modify or eliminate antibody-binding epitopes, alter receptor conformation, and generate soluble decoy isoforms, ultimately reducing effective drug engagement and enabling immune escape [55,56,57]. Understanding tumor-specific splice variants is therefore essential for anticipating, monitoring, and overcoming resistance to antibody-based therapies.

### 4.1. Alternative Splicing as a Driver of Antigenic Plasticity

AS enables tumors to remodel both surface and secreted antigens at the mRNA level, enabling dynamic adaptation and therapeutic resistance. The following four recurrent AS-driven mechanisms contribute to this process:(1)**Loss of Extracellular Epitopes Through Exon Skipping**

The removal or modification of exons that encode extracellular or membrane-proximal domains produces truncated receptors lacking antibody- or chimeric antigen receptor (CAR)-T-recognized regions. These isoforms often fail to reach or present on the cell surface, mimicking genetic antigen loss. AS-mediated epitope deletion allows tumor cells to escape immune detection while retaining minimal intracellular signaling (see Section 4.2.1 and Section 4.2.2).

(2)
**Structural Remodeling of Receptor Conformation**


AS can modify receptor architecture by altering dimerization domains or extracellular folding motifs. Such changes may sterically obscure therapeutic epitopes or disrupt normal receptor accessibility, thereby diminishing antibody–receptor binding affinity and limiting the pharmacologic reach of targeted therapies (see Section 4.2.3).

(3)
**Generation of Soluble Decoy Isoforms**


Exclusion of the transmembrane domain-encoding exons generates secreted isoforms that retain ligand- or antibody-binding potential. These soluble variants act as antibody sinks, sequestering therapeutic antibodies away from tumor cells. This mechanism is particularly evident in immune-checkpoint and angiogenic pathways (see Section 4.2.3 and Section 4.2.4).

(4)
**Signaling Rewiring Through Isoform Switching**


Alterations in cytoplasmic signaling domains or dimerization interfaces can produce isoforms capable of sustaining downstream oncogenic signaling independent of canonical receptor–ligand engagement. These splice variants maintain proliferation and survival signals despite pharmacologic blockades (see Section 4.2.5).

Collectively, these AS-driven mechanisms provide tumors with remarkable antigenic flexibility, contributing to both primary non-response and acquired resistance after initially effective therapy [55,56,57]. Understanding these splicing-mediated alterations provides the conceptual foundation for Section 4.2, which describes how these mechanisms manifest across clinically important therapeutic targets.

### 4.2. Target-Specific Manifestations of Splicing-Mediated Resistance

#### 4.2.1. CD19/CD22: Exon Skipping-Mediated Antigen Loss in B-Cell Malignancies

(1)
**Biological Roles of CD19 and CD22**


CD19 and CD22 are essential B-cell receptors involved in modulating B-cell receptor (BCR) signaling and are widely exploited as therapeutic targets in B-cell acute lymphoblastic leukemia (B-ALL). CD19 amplifies pre-BCR signaling, thereby promoting the proliferation and differentiation of late pre-B cells through the activation of downstream kinases such as PI3K and LYN. In contrast, CD22 dampens BCR-mediated signaling through immunoreceptor tyrosine-based inhibitory motifs (ITIMs) [58,59].

(2)
**Therapeutic Outcomes and Resistance Dynamics**


CD19-directed CAR-T therapy (Figure 2A) and CD22-directed ADCs have revolutionized the treatment for relapsed or refractory B-ALL, achieving high complete remission (CR) rates, even in chemotherapy-refractory patients [50,60,61,62,63]. However, despite these impressive initial responses, relapse remains common. A substantial proportion of patients eventually present with antigen-negative disease, in which leukemic cells no longer express a sufficient therapeutic target [64,65]. For example, while CD19 CAR-T therapy achieves CR rates of approximately 80–90%, more than half of responders relapse within one year, frequently with CD19-negative leukemia [61,65]. Increasing evidence shows that aberrant RNA splicing is a major driver of these antigen-loss relapses.

(3)
**RNA Splicing-Mediated Mechanisms of Resistance**


Sotillo et al. demonstrated that CD19 exon 2 skipping yields truncated isoforms lacking the extracellular epitope recognized by CAR-T cells [61] (Figure 2A). The loss of splicing factors SRSF3 or PTBP1 disrupts exon 2 inclusion, leading to the accumulation of aberrant isoforms that evade CAR-T recognition [61,66].

CD22 undergoes analogous alterations. Zheng et al. reported that relapsed B-ALL frequently harbors CD22 exon 2 or exon 5–6 skipping events. The exon 2 skipping abolishes protein production by eliminating the start codon, while the exon 5-6 skipping produces truncated proteins that lack an antibody-recognition domain, conferring resistance to CD22-directed therapies such as inotuzumab ozogamicin [62].

In summary, these findings establish AS-mediated exon skipping in CD19 and CD22 as a central mechanism of treatment failure in B-cell malignancies (Table 1).

#### 4.2.2. CD20: Splicing-Related Modulation of Antigen Density and Resistance to Anti-CD20 Therapy

(1)
**Biological Roles of CD20**


CD20 (MS4A1) is a tetraspanning membrane protein expressed from the pre-B stage through mature B cells, where it regulates B-cell activation, calcium flux, and the stabilization of BCR signaling [100,101,102]. Its high expression on malignant B cells—but absence on hematopoietic stem cells and plasma cells—makes CD20 an ideal therapeutic target in B-cell malignancies [101,103,104].

(2)
**Therapeutic Outcomes and Resistance Dynamics**


Rituximab, a type I anti-CD20 monoclonal antibody, has transformed treatment for non-Hodgkin lymphoma (NHL) and chronic lymphocytic leukemia (CLL) [105,106]. Subsequent development of second- and third-generation anti-CD20 antibodies (ofatumumab and obinutuzumab) further enhanced CDC, ADCC, and direct cell-death pathways [106,107,108,109].

Despite these therapeutic advances, both primary and acquired resistance remain major clinical challenges [105,110,111]. Resistant tumors commonly display reduced CD20 antigen density or evolve CD20-negative or CD20-dim phenotypes, a well-recognized mechanism of rituximab failure [67,68,69,70]. Additional resistance mechanisms include the upregulation of complement regulatory proteins such as CD55 and CD59, which limit CDC, and alterations in lipid raft composition that disrupt CD20 membrane clustering for optimal antibody crosslinking [71,107,109].

Accumulating evidence indicates that aberrant MS4A1 splicing contributes to these escape mechanisms, with elevated levels of non-canonical transcripts correlating with reduced surface CD20 expression and inferior clinical outcomes [69,72,73].

(3)
**RNA Splicing-Mediated Mechanisms of Resistance**


The AS of MS4A1 generates truncated isoforms lacking exons 3–5, which have been detected in CLL and lymphoma following rituximab exposure [71,72]. These truncated proteins are retained in intracellular compartments, fail to localize to the membrane, markedly reduce effective CD20 density, and thereby impair rituximab binding and weaken CDC and ADCC [68,69,71]. Other splice variants lack the extracellular loops required for antibody recognition yet retain transmembrane segments, enabling them to heterodimerize with wild-type CD20 and act as dominant negative molecules that destabilize surface expression [68,73].

Taken together, CD20 splicing dysregulation represents a clinically meaningful route of immune escape [72,73]. Exon-skipped or truncated isoforms diminish antigen availability and reduce antigen levels, impair effector cell engagement, and diminish the binding of anti-CD20 antibodies. These findings highlight CD20 splicing both as a biomarker of resistance and a potential therapeutic target for corrective splicing strategies.

#### 4.2.3. ERBB Family Receptors: Divergent Splicing-Driven Resistance

The ERBB receptor tyrosine kinase family—which includes EGFR (ERBB1), HER2 (ERBB2), HER3 (ERBB3), and HER4 (ERBB4)—regulates essential processes including proliferation, survival, differentiation, and motility across diverse tissues [77,112]. These receptors share a conserved domain architecture composed of an extracellular ligand-binding domain, a single-pass transmembrane region, and an intracellular kinase domain that activates the MAPK/ERK, PI3K/AKT, and JAK/STAT signaling pathways.

In cancer, ERBB signaling becomes dysregulated through receptor overexpression, activating mutations, or aberrant ligand availability, driving aggressive tumor phenotypes and poor prognosis. These oncogenic features enabled the development of transformative antibody-based therapies that target EGFR or HER2. Despite their structural similarity and overlapping downstream pathways, the mechanisms of resistance between EGFR- and HER2-directed therapies are notably divergent. Emerging evidence reveals that AS contributes to immune evasion and therapeutic escape in both receptors, but through distinct isoforms and biological consequences. The following sections outline these distinct splicing-mediated resistance mechanisms.

##### EGFR: Alternative Splicing and Resistance to Anti-EGFR Therapy

(1)
**Biological Roles of EGFR**


Epidermal growth factor receptor (EGFR/ERBB1) is a ligand-activated receptor tyrosine kinase that undergoes dimerization upon ligand binding, initiating downstream signaling [113,114,115]. EGFR is frequently overexpressed or dysregulated in non-small cell lung cancer (NSCLC), colorectal cancer (CRC), head and neck squamous cell carcinoma (HNSCC), and glioblastoma, where it promotes tumorigenesis and correlates with poor prognosis [113,114,115,116,117].

(2)
**Therapeutic Outcomes and Resistance Dynamics**


Anti-EGFR monoclonal antibodies (cetuximab and panitumumab) improve survival in metastatic CRC and HNSCC by blocking receptor–ligand interactions, preventing EGFR activation, and inducing ADCC [118,119,120].

Despite these benefits, both primary and acquired resistance are pervasive. Established mechanisms include impaired antibody binding, compensatory activation of bypass pathways (e.g., HER2 or MET), and alterations in receptor internalization or antigen presentation [121,122,123].

Emerging evidence indicates that alternative splicing contributes to these resistance patterns. Tumor cells can produce soluble and truncated EGFR isoforms that reduce antibody accessibility or function as ligand/antibody decoys [74,75]. Elevated soluble EGFR (sEGFR) levels have been reported in glioblastoma and CRC, where they correlate with altered responses to anti-EGFR antibodies [74,75,76].

(3)
**RNA Splicing-Mediated Mechanisms of Resistance**


Soluble EGFR (sEGFR), generated through exon 16–17 skipping or the use of alternative terminal exons, retains ligand-binding ability but lacks the transmembrane domain, allowing it to circulate in the extracellular milieu [74]. By binding EGF-family ligands or directly sequestering therapeutic antibodies, sEGFR reduces antibody availability at the tumor surface. Other splice variants delete portions of the extracellular ligand-binding region, modifying epitope exposure or altering receptor conformation. These truncated isoforms may interfere with antibody recognition, promote receptor shedding, or facilitate ligand-independent signaling that bypasses EGFR inhibition [74,124].

Collectively, aberrant EGFR splicing contributes to reduced surface antigen density, increased decoy isoform production, and activation of bypass pathways—driving resistance to anti-EGFR therapies.

##### HER2: Exon 16 Skipping and Receptor Rewiring

(1)
**Biological Roles of HER2**


HER2 (ERBB2) can adopt a constitutively active conformation even in the absence of ligand interaction, driving the persistent activation of downstream signaling [77,112,125].

HER2 amplification or overexpression in breast, gastric, and lung cancers is associated with aggressive tumor behavior and poor clinical outcomes [126,127,128].

(2)
**Therapeutic Outcomes and Resistance Dynamics**


Anti-HER2 monoclonal antibodies (trastuzumab and pertuzumab) and ADCs (Trastuzumab Emtansine; T-DM1 and trastuzumab deruxtecan; T-DXd), have markedly improved outcomes for patients with HER2-positive malignancies [129,130,131,132,133,134,135,136].

Trastuzumab and pertuzumab block HER2 dimerization and disrupt downstream signaling [130,131,137,138], while T-DM1 and T-DXd deliver cytotoxic payloads selectively into HER2-expressing tumor cells [132,133,134,135,136,139,140,141] (Figure 2B). Landmark clinical trials, including CLEOPATRA, DESTINY-Breast03, and DESTINY-Gastric01, have demonstrated meaningful improvement in response rates and survival benefits compared with conventional chemotherapy [130,135,136].

Despite their therapeutic impact, resistance to HER2-targeted therapy frequently develops. Tumors that progress on trastuzumab or T-DM1 often display altered HER2 splicing profiles, including increased expression of Δ16HER2, an alternatively spliced isoform associated with poor therapeutic response [77,78,79,80]. Δ16HER2 has been implicated in resistance to trastuzumab, T-DM1, and T-DXd [79,80]. Furthermore, the heterodimerization of Δ16HER2 with HER3 sustains PI3K/AKT signaling even under dual HER2/HER3 blockade, contributing to pertuzumab resistance [81,82].

(3)
**RNA Splicing-Mediated Mechanisms of Resistance**


Δ16HER2, generated through exon 16 skipping, is the most clinically relevant HER2-spliced isoform linked to treatment resistance [81,83,84]. Δ16HER2 forms constitutively active homodimers that drive MAPK and PI3K/AKT activation, while distorting or partially masking the trastuzumab-binding epitope on domain IV, reducing antibody accessibility and diminishing ADCC. Δ16HER2 also enhances Src kinase signaling, increasing invasiveness and metastatic potential [81,83,84] (Figure 2B). The splicing of HER2 pre-mRNA is tightly regulated by RNA-binding proteins such as SRSF3 and hnRNP H1, which promote exon 16 skipping and increase Δ16HER2 expression [142].

These combined findings suggest that Δ16HER2-driven splicing alterations reshape HER2 signaling and antigen structure, establishing this isoform as a central driver of resistance to HER2-targeted therapies and a promising biomarker for treatment stratification.

#### 4.2.4. VEGF: Isoform Switching and Soluble Ligand Decoys

(1)
**Biological Roles of VEGF**


The vascular endothelial growth factor (VEGF) family (VEGFA/B/C/D, placental growth factor; PlGF) is a central regulator of angiogenesis, orchestrating endothelial cell proliferation, migration, and survival under both physiological and pathological conditions [44,85,143,144]. This family primarily signals through three receptor tyrosine kinases—VEGFR1 (Flt-1), VEGFR2 (KDR), and VEGFR3 (Flt-4) [44,144]. The aberrant activation of the VEGF–VEGFR axis drives tumor angiogenesis, metastasis, and therapeutic resistance [44,85,86,143,144,145].

(2)
**Therapeutic Outcomes and Resistance Dynamics**


Bevacizumab (Avastin), a monoclonal antibody targeting VEGFA, prevents VEGFA binding to VEGFR1/2, thereby suppressing angiogenesis and tumor growth (Figure 2C) [87,143,144,146]. Clinical trials—including AVF2107g (colorectal cancer) and AVOREN (renal cell carcinoma)—have demonstrated significant improvements in overall survival and established bevacizumab-based regimens as a standard of care [87,143,144,147].

Despite these benefits, a subset of patients exhibits intrinsic or acquired resistance. Increasing evidence shows that resistance is mediated by alternative splicing of VEGFA, particularly through upregulation of the anti-angiogenic VEGF165b isoform [45,46,56,86,88]. A pivotal study by Varey et al. demonstrated that tumors with high VEGF165b expression respond poorly to bevacizumab therapy because the antibody is sequestered by this inactive isoform, thereby reducing its availability to neutralize pro-angiogenic VEGF165a and effectively acting as a decoy sink [56,89]. Furthermore, Boudria et al. showed that bevacizumab treatment itself can induce VEGF165b expression, reinforcing resistance and promoting a more invasive tumor phenotype [46].

(3)
**RNA Splicing-Mediated Mechanisms of Resistance**


The VEGFA undergoes alternative 3′ splice-site selection within exon 8, generating pro-angiogenic VEGF-Axxx isoforms (e.g., VEGF165a), including exon 8a, and anti-angiogenic VEGF-Axxxb isoforms (e.g., VEGF165b), including exon 8b [85,86,89]. This splicing event alters the six amino acids at the C-terminus, leading to markedly different receptor activation [85,87,89,90,148].

VEGF165b binds VEGFR2 with comparable affinity to VEGF165a but fails to trigger receptor phosphorylation, thereby inhibiting angiogenic signaling [91,92,93,148]. While VEGF165b predominates to maintain vascular homeostasis in normal tissues, tumors exhibit isoform switching that reduces VEGF165b and elevates VEGF165a expression, shifting the angiogenic balance toward tumor vascularization [89,90] (Figure 2C). VEGFA splicing is controlled by several serine/arginine-rich (SR) proteins, which are phosphorylated by SRPK1/2 and CLK1 kinases [149,150,151,152,153]. Hypoxia and TGF-β signaling promote the nuclear translocation of these SR proteins, enhancing proximal splice-site (exon 8a) usage and upregulating the pro-angiogenic VEGF165a isoform [149,154]. Conversely, inhibitory splicing regulators such as TIA-1 favor distal splice-site inclusion (exon 8b), thereby promoting VEGF165b expression [155].

Overall, VEGFA isoform switching—particularly upregulation of VEGF165b—acts as a soluble ligand decoy that sequesters bevacizumab, representing a major splicing-mediated mechanism of resistance to anti-VEGF therapies.

#### 4.2.5. PD-1/PD-L1: Soluble Isoforms and Immune Checkpoint Blockade Resistance

(1)
**Biological Roles of PD-1 and PD-L1**


Programmed cell death protein (PD-1) and its ligand PD-L1 (CD274) form a key immune checkpoint pathway that downregulates T-cell activation to maintain peripheral tolerance and prevent autoimmunity [156,157].

PD-1 is expressed on activated T cells, B cells, and NK cells, where engagement with PD-L1 recruits SHP2 phosphatase to inhibit T-cell receptor (TCR) and CD28 signaling, suppressing effector cytokine production and promoting T-cell exhaustion [55,94,156,158]. PD-L1 is expressed on antigen-presenting cells and various types of tumors, enabling cancer cells to evade cytotoxic T-cell surveillance through constitutive or inducible checkpoint activation [55,159,160].

(2)
**Therapeutic Outcomes and Resistance Dynamics**


Immune checkpoint inhibitors (ICIs), including anti-PD-1 antibodies (nivolumab and pembrolizumab) and anti-PD-L1 antibodies (atezolizumab), have revolutionized cancer immunotherapy in NSCLC, melanoma, and renal cell carcinoma [158,159,160]. By blocking PD-1/PD-L1 interactions, these agents restore cytotoxic T-cell activity and induce durable responses in a subset of patients (Figure 2D).

However, a significant proportion of patients exhibit primary non-response, and many initial responders ultimately develop acquired resistance [55,95,158,159,160,161]. Growing evidence implicates AS of PD-1 and PD-L1 as a significant, yet underrecognized, mechanism contributing to these failures [95,98,156,157].

Aberrant AS can reduce membrane expression, disrupt antibody-binding epitopes, or generate soluble PD-1 and PD-L1 (sPD-1 and sPD-L1) isoforms. These act as decoy receptors or ligands that sequester therapeutic antibodies and attenuate immune activation, collectively diminishing the efficacy of immune checkpoint blockade and promoting therapeutic resistance [94,95,96].

(3)
**RNA Splicing-Mediated Mechanisms of Resistance**


(**i**) **PD-1 splicing**

The *PDCD1* gene consists of five exons, with exon 3 encoding the transmembrane domain. The skipping of exon 3 produces the ΔEx3 PD-1 isoform (sPD-1), which is secreted rather than membrane-bound. This soluble form competitively binds PD-L1, functioning as a decoy receptor that counteracts inhibitory PD-1 signaling and promotes T-cell activation [94,95,96,97].

Production of ΔEx3 PD-1 is tightly regulated by several splicing factors. HNRNPK binds an exonic splicing silencer (ESS) within exon 3 to suppress ΔEx3 PD-1 formation and maintain membrane-bound PD-1 expression, thereby reinforcing T-cell exhaustion in the tumor microenvironment [96]. MATR3 and DDX5 similarly promote exon 3 inclusion in activated T cells exposed to tumor-derived cytokines [97]. In contrast, inhibition of SPRK1—an upstream regulator of SRSF1 phosphorylation—shifts PD-1 splicing toward ΔEx3 PD-1, reducing surface PD-1 and enhancing anti-tumor immunity [95].

(**ii**) **PD-L1 splicing**

PD-L1 (CD274) also undergoes alternative splicing that generates soluble isoforms lacking the transmembrane domain. The skipping of exons 6 or 7 produces sPD-L1 variants that are secreted and retain the ability to bind PD-1 or therapeutic antibodies, thereby diminishing the efficacy of checkpoint blockade [55,94,98] (Figure 2D). Multiple splice isoforms—including PD-L1v242, PD-L1v229, and PD-L1Δ3— have been detected in NSCLC and breast cancer, where they circulate systemically and sequester anti-PD-L1 antibodies, diminishing therapeutic efficacy [55,98,99].

PD-L1 splicing is regulated by RNA-binding proteins such as TDP-43 and chromatin-associated factors that are dynamically modulated by the tumor microenvironment, particularly under hypoxic or stress conditions [55].

These observations collectively imply that the AS of PD-1 and PD-L1—particularly the generation of sPD-1 and sPD-L1 isoforms—acts as a decoy system that sequesters therapeutic antibodies and blunts immune checkpoint blockade, representing a major splicing-mediated mechanism of resistance to anti-PD-1/PD-L1 immunotherapies.

## 5. Advanced Strategies to Counter Aberrant Regulation of RNA Splicing in Cancer

Aberrant RNA splicing has emerged as a critical driver of resistance to antibody-based cancer therapies. Since tumors exploit splicing plasticity to remodel antigen structure, receptor signaling, and immune interactions, there is an urgent need for next-generation strategies capable of anticipating, detecting, and therapeutically counteracting isoform-driven escape. The sections below highlight several emerging approaches that are reshaping splicing-informed precision oncology.

### 5.1. Comprehensive Mapping of Cancer Splicing Landscapes

High-resolution RNA sequencing and full-length transcriptome profiling are essential for characterizing tumor- and tissue-specific splicing programs. Long-read sequencing technologies [162,163,164] have uncovered numerous clinically actionable isoforms that are often missed by short-read approaches, including structural variants directly linked to therapeutic response. In parallel, large-scale resources such as the ASCancer Atlas [165] systematically catalog splicing alterations across malignancies. These efforts are critical for identifying functional splice variants that drive tumor evolution and therapeutic resistance, ultimately enabling the development of isoform-based biomarkers and therapeutic targets.

### 5.2. Integrating Splicing Signatures into Diagnostics and Patient Monitoring

Incorporating splicing signatures into diagnostic workflows represents a key advancement toward precision oncology. The detection of circulating isoforms—such as sPD-L1 or shifts in VEGF165a/VEGF165b ratios—using liquid biopsy offers a minimally invasive strategy for real-time assessment of treatment responses and the early detection of resistance [166,167]. Integrating splicing information with genomic and proteomic profiling enables adaptive treatment decisions, improving patient stratification and allowing for intervention before clinical relapse occurs.

### 5.3. Therapeutic Targeting of Splicing Regulators

Pharmacologic modulation of splicing regulators —including SR proteins, HNRNPs, and other RNA-binding factors—offers a promising strategy to normalize dysregulated splicing programs. The inhibition of SRPK1, an upstream regulator of SRSF1, enhances ΔEx3 PD-1 formation and improves T-cell activation, whereas targeting PTBP1 has been shown to preserve CD19 expression and mitigate antigen- loss relapse. By reprogramming oncogenic or immune-evasive splice variants, splicing factor-directed therapies may restore sensitivity to antibody-based treatments and expand the therapeutic window for resistant tumors.

### 5.4. Clinical Applications of Antisense Oligonucleotides (ASOs) and RNA Editing

ASO and RNA-editing platforms provide versatile tools for redirecting pathogenic splicing events toward therapeutically favorable isoforms [168,169,170]. Proof-of-concept studies have demonstrated their applicability across diverse therapeutic targets. For example, ASO-mediated correction restores CD22 exon 2 inclusion to overcome resistance to inotuzumab ozogamicin, induces ΔEx3 PD-1 to reinvigorate exhausted T cells [96,157], and shifts splicing toward the sVEGFR2 isoform to inhibit tumor angiogenesis [171]. ASO-based modulation offers a reversible, tunable, and cell type-specific strategy distinct from DNA-targeted approaches.

Parallel advances in RNA-targeting CRISPR (clustered regularly interspaced short palindromic repeats)/Cas13 and RNA base editing technologies further expand opportunities for programmable manipulation of splice-site manipulation without altering genomic DNA [172,173,174]. Cas13 can be guided to pre-mRNA sequences adjacent to splice junctions to sterically block spliceosome assembly or splicing factor binding [173,174,175], thereby inducing or preventing exon inclusion with high specificity. Emerging evidence indicates the promising potential of Cas13-based splice modulation to correct disease-driving splicing defects and modulate therapeutic isoforms in a gene-agnostic approach [176]. These technologies represent an emerging frontier in RNA-level therapeutics that complement ASO-based approaches without the irreversible risks of genome editing.

### 5.5. Development of Isoform-Selective Antibodies and ADCs

Designing antibodies or ADCs that selectively recognize tumor-specific splice variants such as Δ16HER2, VEGF165a, or sPD-L1 offers a direct strategy to overcome isoform-mediated escape. Isoform-restricted epitopes provide opportunities to enhance therapeutic precision, reduce off-target toxicity, and restore efficacy in tumors that evade conventional antibody-based strategies. Isoform-selective biologics represent the logical next step in advancing from pathway-level inhibition toward truly molecularly precise immunotherapy.

## 6. Conclusions and Future Perspectives

Alternative splicing is a highly dynamic post-transcriptional mechanism that profoundly shapes tumor biology by remodeling antigen structure, receptor signaling, and immune interactions. Across major therapeutic targets—including CD19/CD20/CD22, EGFR/HER2, VEGFA, and PD-1/PD-L1—aberrant splicing generates isoforms that remove therapeutic epitopes, alter receptor conformation, or produce soluble decoys. These AS-derived variants undermine the efficacy of monoclonal antibodies, ADCs, and immune checkpoint inhibitors, representing a major and often under-recognized source of therapeutic resistance.

Moving forward, the next stage of precision oncology must explicitly integrate RNA splicing as a core determinant of treatment response. While current clinical paradigms primarily emphasize genomic mutations, splicing alterations are often more dynamic, reversible, and directly linked to treatment sensitivity. We anticipate that isoform profiling will become an integral component of diagnostic evaluation; liquid biopsy-based splicing biomarkers will guide real-time therapeutic adjustments; and RNA-targeted interventions—including ASOs, splice-switching oligonucleotides, and emerging RNA-editing platforms—will be implemented in combination with antibody-based therapies.

Ultimately, overcoming antibody resistance will require a conceptual shift—from exclusively targeting proteins to understanding and therapeutically modulating the RNA processes that generate them. As oncology moves toward deeply mechanistic, RNA-centric precision medicine, splicing-directed strategies offer strong promise for restoring antigen expression, preventing immune escape, and achieving more durable and transformative clinical responses.

## Figures and Tables

**Figure 1 ijms-26-11918-f001:**
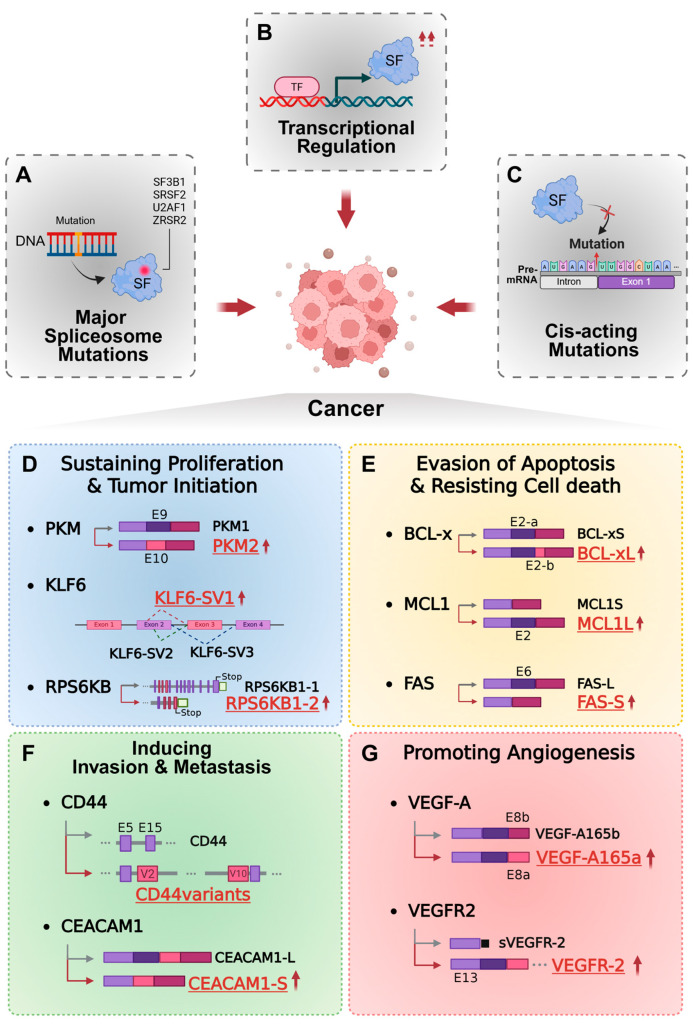
Oncogenic functions of aberrant alternative splicing isoforms in cancer. Aberrant regulation of RNA splicing drives tumor initiation and progression through mutations in spliceosomal components, dysregulated expression of splicing regulators, and cis-acting alterations within splice-site or enhancer elements. Mutations in core spliceosome genes (**A**) (e.g., *SF3B1* and *SRSF2*) impair normal splice-site recognition, while overexpression of splicing factors such as SRSF1 and SRSF3 promotes the production of oncogenic splice variants (**B**). Cis-acting mutations further disrupt splicing factor recruitment, generating tumor-specific isoforms with distinct biological activities (**C**). These aberrant splicing events collectively contribute to multiple cancer hallmarks: enhanced proliferation via PKM2, KLF6-SV1, and RPS6KB1-2 (**D**); resistance to apoptosis through anti-apoptotic isoforms such as Bcl-xL, MCL1L, and soluble Fas-S (**E**); increased invasion and metastasis mediated by CD44v and CEACAM1-S (**F**); and angiogenic activation through isoform switching between VEGF165a and VEGF165b (**G**). Together, these splicing-driven alterations reprogram cellular signaling networks, enabling sustained growth, apoptotic evasion, and metastatic progression that underpin therapeutic resistance.

**Figure 2 ijms-26-11918-f002:**
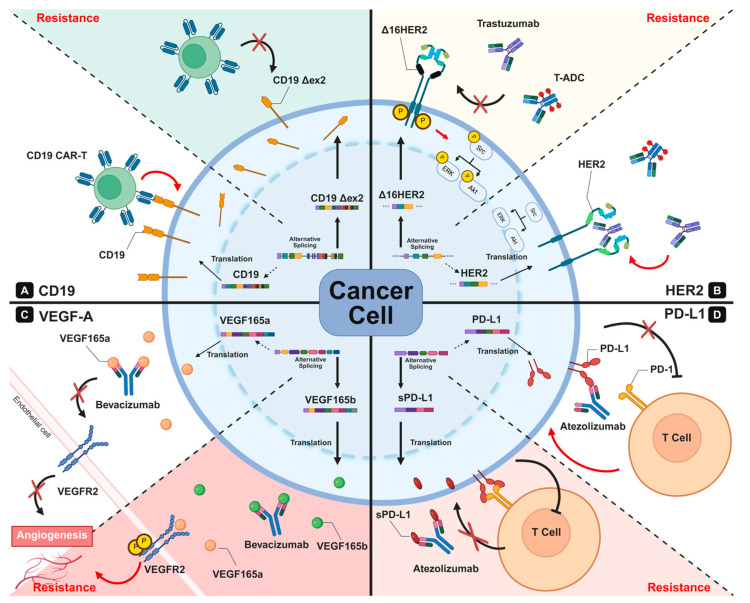
Alternative splicing-mediated mechanisms of resistance to antibody-based therapies. Alternative RNA splicing generates distinct isoforms of therapeutic target proteins, leading to structural and functional alterations that impair antibody recognition and therapeutic efficacy. (**A**) In B-cell malignancies, exon 2 skipping in CD19 produces truncated proteins lacking antibody- or CAR-T-binding epitopes, resulting in immune evasion and relapse after targeted therapy. (**B**) In HER2-positive tumors, the Δ16HER2 isoform that is generated by exon 16 skipping forms constitutively active homodimers that sustain oncogenic signaling and reduce trastuzumab accessibility. (**C**) In VEGF-driven tumors, alternative splicing upregulates VEGF165b, which sequesters bevacizumab as a decoy ligand without blocking VEGFR2 signaling, thereby attenuating anti-angiogenic efficacy and promoting therapeutic resistance. (**D**) In immune checkpoint blockade, alternative splicing of CD274 (PD-L1) produces soluble PD-L1 variants that act as decoy ligands, binding therapeutic antibodies and suppressing T-cell activation.

**Table 1 ijms-26-11918-t001:** Alternative Splicing Events Associated with Resistance to Antibody-Based Therapies.

Targets	Gene Names	Splicing Alterations	Pathogenic ISOFORMS	Mechanism	Clinical Relevance	Reference
**CD19**	*CD19*	Exon 2 skipping	Δex2 CD19	Loss of extracellular epitope → CAR-T recognition failure	CD19^−^ relapse after CAR-T therapy	[61,64,65,66]
**CD22**	*CD22*	(1) Exon 2 skipping (2) Exon 5–6 skipping	(1) Δex2 CD22 (2) Δex5–6 CD22	(1) Loss of protein expression (2) Loss of extracellular domain → ADC/CAR-T escape	- Resistance to inotuzumab ozogamicin - CD22 CAR-T relapse	[62,63]
**CD20**	*MS4A1*	Eson 3–5 skipping	truncated CD20	Fail to localize to membrane → Reduce CD20 density	- Reduced rituximab binding - primary/relapsed CD20-dim lymphoma	[67,68,69,70,71,72,73]
**EGFR**	*ERBB1/* *EGFR*	(1) Exon 16–17 skipping (2) alternative terminal exon usage	soluble EGFR	Loss of TM domain → soluble decoy that binds ligands and antibodies → reduced membrane EGFR availability	- Variable response to anti-cetuximab, panitumumab - potential mechanism of intrinsic resistance	[74,75,76]
**HER2**	*ERBB2*	Exon 16 skipping	Δ16HER2	Constitutive dimerization → distorted trastuzumab epitope → reduced ADCC and binding	Resistance to trastuzumab, T-DM1, T-DXd	[77,78,79,80,81,82,83,84]
**VEGFA**	*VEGFA*	Exon 8a → 8b switching	VEGF165b	Pro- vs anti-angiogenic switch → VEGF165b binds bevacizumab	- Predictor of bevacizumab response - Tumor angiogenesis regulation	[45,46,56,85,86,87,88,89,90,91,92,93]
**PD-1**	PDCD1	Exon 3 skipping	soluble PD-1	Loss of TM domain → soluble decoy binding to PD-L1 → weakens checkpoint blockade	Alter response to anti-PD-1 therapy	[94,95,96,97]
**PD-L1**	CD274	(1) Exon 6 or 7 skipping (2) multiple variants	(1) soluble PD-L1 (2) PD-L1v242, PD-L1v229, PD-L1Δ3	- Secreted isoforms sequester anti-PD-L1 antibodies - Reduced membrane PD-L1	Anti-PD-L1 immunotherapy resistance	[57,94,95,96,98,99]

## Data Availability

No new data were created or analyzed in this study. Data sharing is not applicable to this article.

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
