# Peer review of "Alternative Splicing-Mediated Resistance to Antibody-Based Therapies: Mechanisms and Emerging Therapeutic Strategies"

_ijms, 2025, doi:10.3390/ijms262411918_

Round 1
Reviewer 1 Report
Comments and Suggestions for Authors
The manuscript entitled "Alternative Splicing-Mediated Resistance to Antibody-Based Therapies: Mechanisms and Emerging Therapeutic Strategies" provides a relatively systematic description of aberrant RNA splicing regulation in cancer and its potential contribution to resistance to antibody-based therapeutics. The authors list several commonly used monoclonal antibodies and discuss how RNA splicing aberrations may influence their efficacy. While the topic is timely and relevant, several major issues need to be addressed to enhance the clarity, comprehensiveness, and impact of the manuscript.
- Page 1, line 32: The heading “1. Introduction – Aberrant Regulation of RNA Splicing in Cancer” is potentially misleading. This section primarily focuses on aberrant RNA splicing in Cancer rather than serving as a full introduction. The authors should revise the title to reflect the content accurately and provide a separate, broader Introduction section that contextualizes the entire manuscript for the reader.
- Figures are included in the manuscript but are not cited in the main text. All figures should be referenced at an appropriate point. For Figure 1, the upper panel appears to summarize “Aberrant Regulation of RNA Splicing in Cancer.” If so, this should be explicitly cited in the corresponding text. And the current figure does not clearly illustrate the three major regulatory mechanisms of RNA splicing aberrations. The figure should be revised for clarity and accuracy.
- The manuscript currently discusses four monoclonal antibody targets: CD19/CD22, HER2, VEGF, and PD-1/PD-L1. To comprehensively cover the most clinically successful antibody targets, the authors should also include EGFR, CD20, and CTLA-4.
- Page 3, Section 3 “Antibody-Based Therapeutics and the Impact of RNA Splicing on its Resistance” mainly describes antibody-based therapeutics. Discussion of RNA splicing contributing to therapeutic resistance is limited and presented only briefly at the end. For each antibody-based therapy, explicitly describe how RNA splicing aberrations impact resistance. Additionally, a summary table could be provided to highlight RNA splicing aberrations associated with resistance for each specific therapy, which would enhance clarity and utility.
- Page 4, Section 4 “Mechanisms of Resistance to Antibody-Based Therapies” largely describes the efficacy and mechanisms of action of each antibody. Condense or integrate this section while emphasizing the contribution of RNA splicing aberrations to therapeutic resistance. Detailed descriptions of drug mechanisms may not need to be a standalone section.
- Page 10, Section 5 “Conclusion and Future Perspectives” does not adequately highlight the authors’ perspectives. Sections 5.1 and 5.2 could be moved prior to the conclusion and presented as a separate section, e.g., “Advanced Strategies to Counter Aberrant Regulation of RNA Splicing in Cancer,” to better emphasize potential future directions and novel strategies.
Addressing these points would substantially enhance the manuscript’s clarity, rigor, and impact, and make it suitable for publication in this journal.
Author Response
Response to Reviewers
We sincerely thank the reviewers for their constructive and insightful comments. We have carefully revised the manuscript to improve clarity, accuracy, logical flow, and scientific depth. Below, we provide a point-by-point response to each comment and indicate the corresponding revisions in the manuscript.
[Reviewer 1]
Comment 1.
- Page 1, line 32: The heading “1. Introduction – Aberrant Regulation of RNA Splicing in Cancer” is potentially misleading. This section primarily focuses on aberrant RNA splicing in Cancer rather than serving as a full introduction. The authors should revise the title to reflect the content accurately and provide a separate, broader Introduction section that contextualizes the entire manuscript for the reader.
Response:
We thank the reviewer for this insightful comment. We agree that the previous version of the Introduction was too narrowly focused on cancer-associated aberrant splicing, which could be misleading.
We have fully reorganized the introductory sections as follows:
- A new “Section 1. Introduction” has been added, providing a broader overview of RNA splicing, its biological significance, and its relevance across a wide range of human diseases—not limited to cancer.
- The previous mechanistic content has been moved and expanded into “Section 2. Mechanisms of Aberrant Regulation of RNA Splicing in Cancer” to clearly delineate general principles from cancer-specific dysregulation.
These revisions improve conceptual clarity, enhance the logical flow, and provide the manuscript with the structure expected for a comprehensive review article.
Comment 2.
- Figures are included in the manuscript but are not cited in the main text. All figures should be referenced at an appropriate point. For Figure 1, the upper panel appears to summarize “Aberrant Regulation of RNA Splicing in Cancer.” If so, this should be explicitly cited in the corresponding text. And the current figure does not clearly illustrate the three major regulatory mechanisms of RNA splicing aberrations. The figure should be revised for clarity and accuracy.
Response:
We thank the reviewer for this helpful comment. In the revised manuscript, both figures are now explicitly cited at appropriate points in the main text:
- Figure 1 is cited in Section 2, where three mechanisms of aberrant regulation of RNA splicing in cancer and Section 3, where the four major oncogenic consequences of aberrant splicing are described.
- Figure 2 is cited in Section 4, in the description of splicing-mediated mechanisms of resistance to antibody-based therapies.
In addition, Figure 1 has been redrawn for clarity. The upper panel is now divided into three clearly delineated mechanistic categories:
- Spliceosomal mutations;
- Altered expression of splicing regulators;
- Cis-regulatory element alterations.
We also added panel labels and expanded the figure legend to improve interpretability. These revisions ensure that the figures more accurately reflect the textual content and enhance readability.
Comment 3.
- The manuscript currently discusses four monoclonal antibody targets: CD19/CD22, HER2, VEGF, and PD-1/PD-L1. To comprehensively cover the most clinically successful antibody targets, the authors should also include EGFR, CD20, and CTLA-4.
Response:
This suggestion is highly appreciated. In accordance with this recommendation, the manuscript now includes two newly added subsections addressing additional antibody targets:
- 2.2. CD20: Splicing-Related Modulation of Antigen Density and Resistance to Anti-CD20 Therapy;
- 2.3.1. EGFR: Alternative Splicing and Resistance to Anti-EGFR Therapy.
We also evaluated whether a subsection on CTLA-4 should be incorporated. After reviewing the available literature, we determined that soluble CTLA-4 (sCTLA-4) has context-dependent and sometimes contradictory immunological effects. Some studies report associations between high sCTLA-4 levels and improved checkpoint blockade responses, whereas others link elevated sCTLA-4 to treatment resistance. Given that these findings vary by tumor type and lack a unified mechanistic explanation, including CTLA-4 would risk introducing ambiguity and weakening the mechanistic focus of the review.
To maintain a clear and cohesive narrative centered on well-established splicing events that directly drive antibody resistance, the manuscript focuses on targets with robust mechanistic support—CD19, CD20, CD22, EGFR, HER2, VEGF, and PD-1/PD-L1.
Comment 4.
- Page 3, Section 3 “Antibody-Based Therapeutics and the Impact of RNA Splicing on its Resistance” mainly describes antibody-based therapeutics. Discussion of RNA splicing contributing to therapeutic resistance is limited and presented only briefly at the end. For each antibody-based therapy, explicitly describe how RNA splicing aberrations impact resistance. Additionally, a summary table could be provided to highlight RNA splicing aberrations associated with resistance for each specific therapy, which would enhance clarity and utility.
Response:
We fully agree with the reviewer’s comment. In the revised manuscript, the previous extensive description of antibody-based therapeutics has been significantly condensed, and the entire section has been reorganized to improve clarity and readability. Specifically,
- Section 4.1 (previously Section 3) now provides a streamlined, conceptual overview of the four major mechanisms by which alternative splicing drives resistance to antibody-based therapies.
- Section 4.2 has been restructured into target-specific subsections (CD19, CD20, CD22, EGFR, HER2, VEGF, PD-1/PD-L1), allowing readers to clearly follow how splicing alterations affect each therapeutic pathway.
- In alignment with the reviewer’s recommendation, we added Table 1, which systematically summarizes all reported alternative splicing events linked to antibody resistance, including exon changes, isoform names, and clinical relevance etc.,.
These revisions substantially enhance the organization, coherence, and accessibility of the manuscript.
Comment 5.
- Page 4, Section 4 “Mechanisms of Resistance to Antibody-Based Therapies” largely describes the efficacy and mechanisms of action of each antibody. Condense or integrate this section while emphasizing the contribution of RNA splicing aberrations to therapeutic resistance. Detailed descriptions of drug mechanisms may not need to be a standalone section.
Response:
We have streamlined and harmonized the descriptions of antibody mechanisms across all therapeutic target subsections. Each target-specific section now follows a unified structure consisting of (i) biological roles, (ii) therapeutic outcomes and resistance dynamics, and (iii) RNA splicing–mediated resistance—with the third component substantially expanded to emphasize splicing-driven mechanisms. This restructuring improves clarity, removes redundancy, and aligns the manuscript with the reviewer’s recommendation for a more focused and mechanistically centered presentation.
Comment 6.
- Page 10, Section 5 “Conclusion and Future Perspectives” does not adequately highlight the authors’ perspectives. Sections 5.1 and 5.2 could be moved prior to the conclusion and presented as a separate section, e.g., “Advanced Strategies to Counter Aberrant Regulation of RNA Splicing in Cancer,” to better emphasize potential future directions and novel strategies.
Response:
We appreciate the reviewer’s structural recommendation and have implemented it accordingly. Specifically,
- Section 5 has been renamed “Advanced Strategies to Counter Aberrant Regulation of RNA Splicing in Cancer” to clearly reflect its forward-looking focus.
- The former subsections 5.1–5.2 have been reorganized under this section to consolidate and emphasize emerging directions in splicing-targeted therapeutics.
- Section 6 (“Conclusion and Future Perspectives”) has been rewritten to succinctly articulate the authors’ perspective on the next phase of splicing-informed oncology and its clinical implications.
These revisions increase coherence and provide a more future-oriented framework consistent with the reviewer’s suggestion.
Reviewer 2 Report
Comments and Suggestions for Authors
This review provides a comprehensive overview of RNA alternative splicing (AS)-mediated resistance to antibody-based cancer therapies. The topic is timely, the structure is largely clear, and the literature coverage is extensive. Nevertheless, improvements in logical flow, figure integration, and depth of future perspectives are recommended.
- The three mechanistic categories in Introduction §3 would benefit from a schematic (proposed Fig. 0) to illustrate the mutation–regulator–cis-element hierarchy.
- The future perspectives focus on ASOs; please add a paragraph on emerging CRISPR-Cas13 and RNA base-editing studies for splice correction (cite at least two 2024 Nature papers).
- A missing “Table 1” is suggested: a summary table of reported resistance-associated splice events (gene, exon change, detection method, clinical association, reference) as supplementary data.
Author Response
Response to Reviewers
We sincerely thank the reviewers for their constructive and insightful comments. We have carefully revised the manuscript to improve clarity, accuracy, logical flow, and scientific depth. Below, we provide a point-by-point response to each comment and indicate the corresponding revisions in the manuscript.
[Reviewer 2]
Comment 1.
- The three mechanistic categories in Introduction §3 would benefit from a schematic (proposed Fig. 0) to illustrate the mutation–regulator–cis-element hierarchy.
Response:
We have incorporated this recommendation by fully redrawing Figure 1, which now clearly illustrates the hierarchical relationships among spliceosomal mutations, dysregulated splicing regulators, and cis‑acting element alterations. This schematic is placed directly after the relevant text for better integration.
Comment 2.
- The future perspectives focus on ASOs; please add a paragraph on emerging CRISPR-Cas13 and RNA base-editing studies for splice correction (cite at least two 2024 Nature papers).
Response:
An expanded subsection, 5.4, “Clinical Applications of Antisense Oligonucleotides (ASOs) and RNA Editing,” has been incorporated into the manuscript. This section provides a detailed discussion of
- CRISPR/Cas13 and RNA base-editing technologies, including recent advances reported in high-impact journals (2024 Nature papers);
- Splice-correction studies using CRISPR-based platforms.
These additions directly address the reviewer’s request and strengthen the manuscript’s coverage of emerging RNA-targeted therapeutic strategies.
Comment 3.
- A missing “Table 1” is suggested: a summary table of reported resistance-associated splice events (gene, exon change, detection method, clinical association, reference) as supplementary data.
Response:
We have created Table 1, summarizing clinically relevant splicing events associated with each therapeutic target, including gene names, splicing alterations, pathogenic isoforms, mechanism, and their clinical relevance.
Regarding the reviewer’s suggestion to also include detection methods, we chose not to incorporate this column into the final table. Detection approaches for splice variants are highly heterogeneous and often depend on:
- the specific isoform structure (e.g., presence/absence of transmembrane or extracellular domains),
- antibody availability, and
- platform-specific assay design (e.g., RT-PCR primer sets, isoform-specific antibodies, RNA-seq pipelines).
Because many detection methods are widely shared across different targets (e.g., isoform-specific RT-PCR, RNA-seq, Western blot), inclusion in a table would lead to substantial redundancy. Moreover, in some cases, multiple unrelated methods are used for the same splice event, making a single “representative” method misleading.
For clarity and accuracy, we therefore opted to describe detection strategies within the corresponding text sections rather than tabulate them.
Round 2
Reviewer 1 Report
Comments and Suggestions for Authors
Thank you for the revised manuscript. I have carefully reviewed the authors’ responses and the updated version. The authors have adequately addressed all of my previous comments, and the revisions have substantially improved the clarity and quality of the manuscript. However, please double-check the legend of Figure 2, as there appears to be a small error.
Aside from this minor issue, I have no further concerns and recommend the manuscript for publication in its current form.